# Hippocampal activation is associated with longitudinal amyloid accumulation and cognitive decline

Stephanie L Leal[1]*, Susan M Landau[1,2], Rachel K Bell[1], William J Jagust[1,2]*

[1]Helen Wills Neuroscience Institute, University of California, Berkeley, United States; [2]Molecular Biophysics and Integrated Bioimaging, Lawrence Berkeley National Laboratory, Berkeley, United States

**Abstract** The amyloid hypothesis suggests that beta-amyloid (A$\beta$) deposition leads to alterations in neural function and ultimately to cognitive decline in Alzheimer's disease. However, factors that underlie A$\beta$ deposition are incompletely understood. One proposed model suggests that synaptic activity leads to increased A$\beta$ deposition. More specifically, hyperactivity in the hippocampus may be detrimental and could be one factor that drives A$\beta$ deposition. To test this model, we examined the relationship between hippocampal activity during a memory task using fMRI and subsequent longitudinal change in A$\beta$ using PIB-PET imaging in cognitively normal older adults. We found that greater hippocampal activation at baseline was associated with increased A$\beta$ accumulation. Furthermore, increasing A$\beta$ accumulation mediated the influence of hippocampal activation on declining memory performance, demonstrating a crucial role of A$\beta$ in linking hippocampal activation and memory. These findings support a model linking increased hippocampal activation to subsequent A$\beta$ deposition and cognitive decline.

**\*For correspondence:**
stephanieleal@berkeley.edu (SLL);
jagust@berkeley.edu (WJJ)

**Competing interests:** The authors declare that no competing interests exist.

## Introduction

Beta-amyloid (A$\beta$) is one of the core pathologies of Alzheimer's disease (AD) and A$\beta$ deposition is posited to be the earliest event in a long and complex pathological cascade leading to AD. The amyloid cascade hypothesis suggests that soluble forms of A$\beta$ represent the start of AD, which is ultimately associated with damage in synaptic structure and function and the formation of A$\beta$ plaques (*Hardy, 2009*). However, it remains unknown why A$\beta$ deposition occurs in the first place.

A substantial body of preclinical research suggests that synaptic activity is linked to A$\beta$ deposition (*Nitsch et al., 1993*; *Kamenetz et al., 2003*; *Cirrito et al., 2005*; *Bero et al., 2011*). Studies using in vivo microdialysis with concurrent field potential recordings have shown that A$\beta$ levels are directly influenced by synaptic activity, in which increases in neuronal activity are sufficient to modulate A$\beta$ levels and plaque growth in a local, region-specific manner (*Cirrito et al., 2005*; *Bero et al., 2011*). Furthermore, using an acute brain slice model, rapid effects of synaptic activity on A$\beta$ levels have been shown to be primarily linked to synaptic vesicle exocytosis (*Cirrito et al., 2005*). There is evidence that hyperactivity in the CA3 subregion of the hippocampus in aged rodents may underlie memory deficits associated with aging (*Wilson et al., 2005*; *El-Hayek et al., 2013*; *Haberman et al., 2013*). This hyperactivity has been regarded as epileptiform in nature in rodents (*Koh et al., 2010*) and AD transgenic mouse models (*Palop et al., 2007*). A reversal of memory impairment has been shown with a reduction of hippocampal hyperactivity (via a low-dose of an anti-epileptic drug) in age-impaired rodents, AD rodent models, and in mild cognitive impairment (MCI) (*Koh et al., 2010*; *Bakker et al., 2012, 2015*; *Sanchez et al., 2012*). There are at least two proposed hypotheses regarding increased hippocampal activation. It has been suggested that

**eLife digest** Dementia refers to the loss of intellectual or thinking abilities and Alzheimer's disease is its most common cause. Although we don't understand the cause of Alzheimer's disease, we know that at early stages of the disease a protein known as beta-amyloid becomes deposited in the brain in the form of plaques. Evidence suggests that this beta-amyloid is harmful to brain cells. Although we don't know why amyloid is deposited, research in animal models suggests that the activity of the brain cells themselves may lead to its deposition.

We can measure brain activity in living people using a technique known as functional magnetic resonance imaging (fMRI), and we can also measure amyloid deposition using positron emission tomography (PET scanning). Leal et al. have now used these techniques to learn about how brain activity might influence amyloid deposition.

Volunteers initially performed a memory task while their brain activity was measured using fMRI. This gave a "baseline" level of brain activity. Over the course of several years, the volunteers returned for PET scans and further memory tests. Cognitively normal older adults with greater baseline levels of brain activity – particularly in the hippocampus, a brain region involved in the formation of new memories – showed more beta-amyloid accumulation over the next three to four years. Furthermore, people who accumulated more beta-amyloid also showed a more severe decline in memory.

To strengthen these results, a follow up study should be performed that examines how brain activity and amyloid deposition change together over time. In addition, it will be important to test whether methods that reduce brain activity could affect amyloid deposition, thus perhaps reducing the risk of Alzheimer's disease.

hyperactivity within the hippocampus may occur over the course of normal aging, but hyperactivity may eventually lead to the secretion of $A\beta$ more quickly than would otherwise occur. Alternatively, some have suggested that $A\beta$ may lead to increased neuronal activity. These explanations are not mutually exclusive; both may be occurring leading to a vicious cycle of $A\beta$ production.

Human studies have also shown that increased hippocampal activity is associated with memory function and $A\beta$ deposition in aging and MCI (*Bakker et al., 2012*; *Brody et al., 2008*; *Mormino et al., 2012*; *Yassa, 2011*; *Dickerson et al., 2005*). High-resolution fMRI during memory task performance has localized hyperactivity to the dentate gyrus/CA3 subregions of the hippocampus, which has also been linked to memory deficits in aging and MCI (*Bakker et al., 2012*; *Yassa, 2011*). Cognitively normal older adults with high $A\beta$ have increased hippocampal and default mode network (DMN) activation during episodic memory processing (*Mormino et al., 2012*; *Sperling et al., 2009*; *Vannini et al., 2012*). Subjects with MCI have shown heightened medial temporal lobe (MTL) activation during episodic memory processing that is associated with subsequent cognitive decline (*Dickerson et al., 2005*, *2004*; *O'Brien et al., 2010*). A recent study found that individuals with MCI who were $A\beta+$ at baseline showed increased hippocampal activation at baseline and longitudinally in addition to longitudinal cognitive deficits compared to MCI subjects who were $A\beta-$ (*Huijbers et al., 2015*). Apolipoprotein E4 (ApoE4) carriers have also shown increased MTL activation (*Dennis et al., 2010*; *Bookheimer et al., 2000*). Eventually, it appears that hippocampal activity decreases as AD pathology progresses (*O'Brien et al., 2010*; *Celone et al., 2006*).

In addition to MTL activity, human imaging studies have also shown increased cortical activity in regions associated with the DMN, but also in regions beyond the DMN (*Mormino et al., 2012*; *Bookheimer et al., 2000*; *Elman et al., 2014*; *Jones et al., 2016*; *Myers et al., 2014*). A recent study found that older adults with $A\beta$ deposition had increased memory task-related neural activation in the parietal and occipital cortex that was linked to more detailed memory encoding (*Elman et al., 2014*), suggesting the activation increases were compensatory. Other studies have discovered links between $A\beta$ and the DMN such that $A\beta$ aggregates in areas of high intrinsic connectivity (*Myers et al., 2014*) and that there is high connectivity between the posterior DMN and hubs of high frontal lobe connectivity that are associated with $A\beta$ accumulation (*Jones et al., 2016*). High $A\beta$ deposition has been found in locations of cortical hubs, such as posterior cingulate, lateral

temporal, lateral parietal, and medial prefrontal cortices (*Buckner et al., 2009*). It remains unclear as to whether local amyloid deposition is driving local hyperactivity, if region-specific hyperactivity is linked to global amyloid deposition, or both may be occurring. Previous studies have shown that hyperactive neurons were typically found near amyloid plaques in the cortex (*Busche et al., 2008*), however, increased hippocampal activity occurred before transgenic mice began to deposit amyloid plaques. Hippocampal hyperactivity can occur independent of plaque deposition, while increased activity in cortical neurons is more directly linked to the presence of amyloid plaques. This may be due to a difference in local concentrations of amyloid, different vulnerabilities of cortical versus hippocampal neurons, or both (*Busche et al., 2012*). Another possibility is that hippocampal hyperactivity may drive increased cortical activity in downstream regions that are more directly linked to amyloid deposition or that hippocampal hyperactivity is driving tau deposition in the MTL, which then leads to A$\beta$ deposition.

One proposed model suggests that life-long patterns of neural activity may lead to A$\beta$ deposition (*Jagust and Mormino, 2011*). Thus far, most studies have examined the link between A$\beta$ and brain activation using cross-sectional data. Longitudinal studies are essential for testing hypotheses about brain activation leading to A$\beta$ accumulation over time and to determine if the rate of A$\beta$ deposition is different in people who show more or less brain activation.

In the present study, our goal was to examine how hippocampal neural activity influenced the longitudinal accumulation of global A$\beta$ and if these factors were related to changes in cognition. We hypothesized that cognitively normal older individuals with greater hippocampal activity at baseline would show increased longitudinal A$\beta$ accumulation globally and that this would be associated with memory decline. To test this hypothesis, we performed a study with a baseline assessment of memory during acquisition of functional magnetic resonance imaging (fMRI) to measure task-related neural activity, followed by longitudinal [$^{11}$C] Pittsburgh Compound-B positron emission tomography (PIB-PET) imaging to measure global A$\beta$ deposition across time in addition to repeated assessments of memory. We focused on the California Verbal Learning Test Long-Delay Free Recall (CVLT) to measure long-term memory changes over time.

## Results

Forty-five cognitively normal older adults participated in the fMRI experiment, the results of which have been previously published (*Mormino et al., 2012*). For the current study, we included all subjects who had: (1) participated in the original fMRI experiment, (2) at least two PIB scans (first near the time of fMRI), and (3) at least three neuropsychological testing sessions. Fifteen of the 45 subjects did not return for follow-up PIB-PET imaging after their initial PIB scan. Three additional participants were excluded due to technical factors. This left us with a sample of twenty-seven subjects (N = 27) who met the criteria for the study (see *Table 1*).

Participants performed a memory task inside the MRI scanner that involved being shown two hundred images of natural outdoor scenes with instructions to indicate whether water was present in each image. After the MRI scan (15 min later), a surprise recognition task was conducted that

**Table 1.** Participant demographics (N = 27).

| Variable | Mean | Range |
|---|---|---|
| Age (at MRI) (SD) | 76.5 (5.6) | 67–91 |
| Sex (M: F) | 8: 19 | |
| Education, years (SD) | 17.4 (1.8) | 14–20 |
| ApoE4 carriers, n (%) | 8 (30) | |
| Time since MRI (PIB), years (SD) | 3.4 (2.1) | 2.8–6 |
| Time since MRI (CVLT), years (SD) | 2.7 (1.1) | 1.5–6.1 |
| Two PIB scans, n | 19 | |
| Three PIB scans, n | 8 | |

*Key:* MRI = Magnetic resonance imaging, SD = Standard deviation, PIB = Pittsburgh Compound B, CVLT = California Verbal Learning Test.

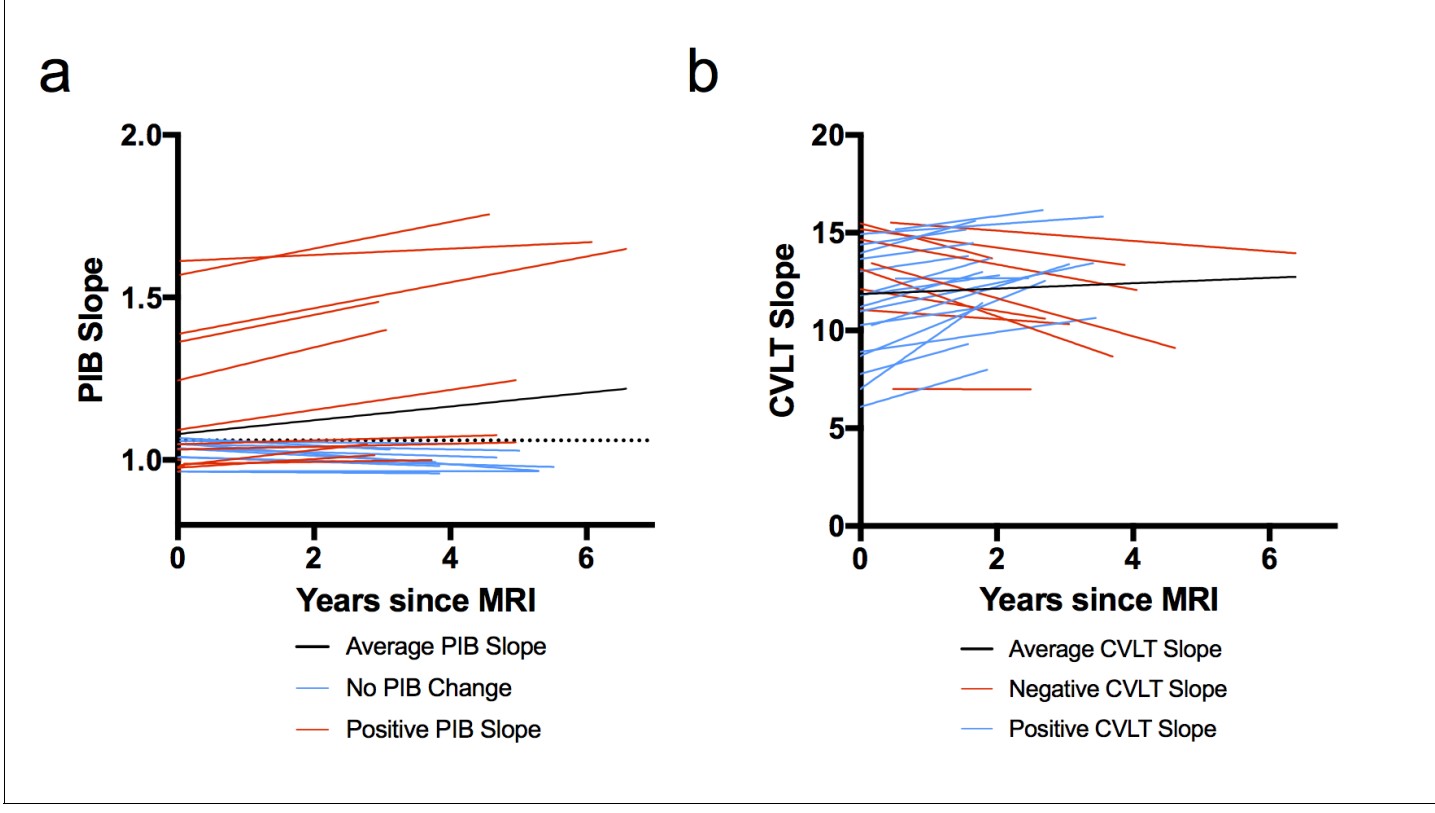

**Figure 1.** Model-derived individual slopes for amyloid and memory change over time. (A) [11C] Pittsburgh Compound-B distribution volume ratio slope (PIB slope) for each participant over time, relative to baseline MRI (x = 0), where black line indicates average PIB slope, blue lines indicate little/no change in PIB, and red lines indicate positive PIB slope. The horizontal dotted line indicates the criterion for a positive PIB scan (1.06), (B) California Verbal Learning Test Long-Delay Free Recall slope (CVLT slope) for each participant over time, relative to baseline MRI (x = 0), where black line indicates average CVLT slope, red lines indicate negative CVLT slope, and blue lines indicate positive CVLT slope. All slopes were obtained from the linear mixed model. The following source data (*Figure 1—source data 1*) is relevant for *Figures 1–4*.

The following source data is available for figure 1:

**Source data 1.** This file contains the source data for *Figures 1–4*.

included all stimuli presented during encoding as well as one-hundred novel stimuli. For each image, subjects were asked if they had seen the image before and were allowed to respond with one of four responses: (1) high-confidence yes, (2) low-confidence yes, (3) high-confidence no, and (4) low-confidence no. The recognition task was self-paced and subjects were encouraged to be as accurate as possible.

We examined three key measures: (1) task-related activity in the right hippocampus and several non-hippocampal regions that showed increased activation in PIB+ compared to PIB- individuals during high-confidence hits versus misses (*Mormino et al., 2012*), (2) PIB distribution volume ratios (DVR) to measure global A$\beta$ deposition across frontal, parietal, temporal, and cingulate cortices, and (3) performance on the CVLT Long-Delay Free Recall to measure changes in long-term memory. The right hippocampus was chosen for analysis because this hippocampal region demonstrated the strongest effect of A$\beta$ on activation in the original study, perhaps because the stimuli were visually presented scenes (*Mormino et al., 2012*).

To examine relationships between hippocampal activation at baseline, longitudinal A$\beta$ accumulation, and longitudinal memory performance, we conducted a series of linear mixed models. All models included age, sex, and education as covariates, as well as a random intercept to account for individual variability in initial PIB DVR or CVLT scores and random slopes to account for individual variability in PIB DVR or CVLT slopes. Age was mean-centered at 76.5 years, education was mean-

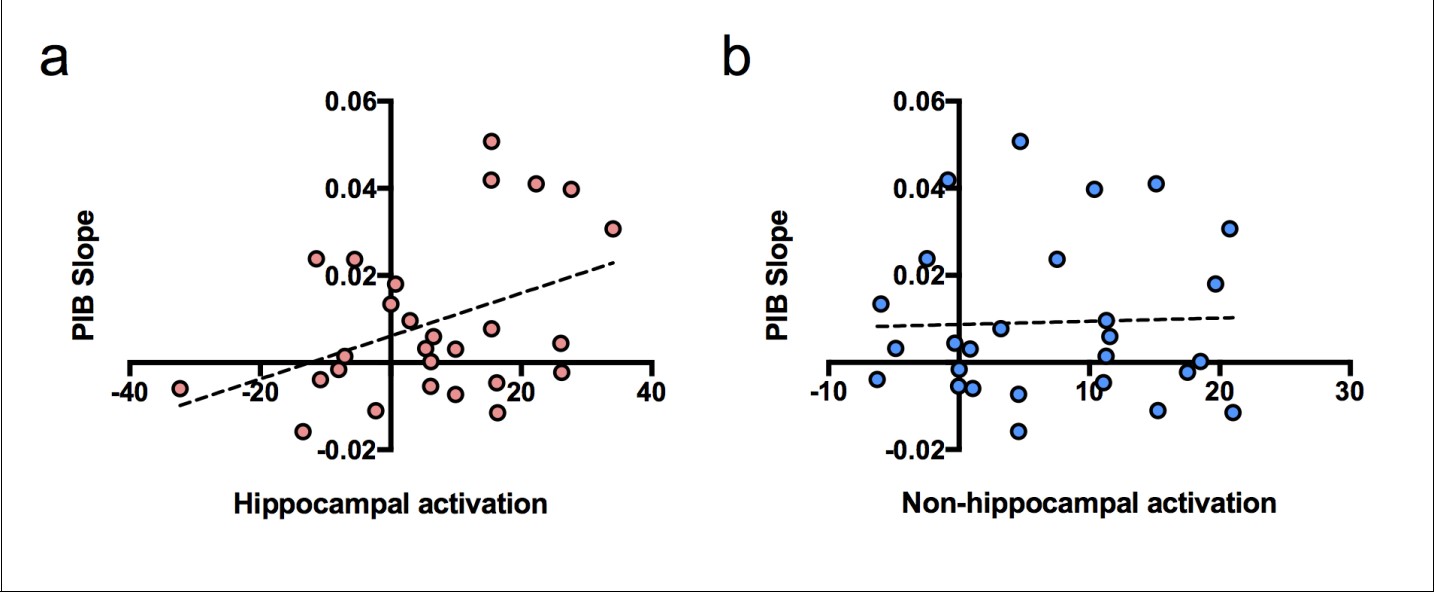

**Figure 2.** Relationship between brain activation at baseline and longitudinal amyloid accumulation. (**a**) Positive relationship between hippocampal activation (contrast values for subsequent hits versus misses) at baseline and [11C] Pittsburgh Compound-B distribution volume ratio (DVR) slopes (PIB slope measured as PIB DVR values over time) obtained from the linear mixed model, (**b**) No relationship between non-hippocampal activation (contrast values for subsequent hits versus misses in inferior frontal cortex and occipital cortex) at baseline and [11C] Pittsburgh Compound-B distribution volume ratio (DVR) slopes (PIB slope measured as PIB DVR values over time) obtained from the linear mixed model.

centered at 17.4 years, and females were the reference group, so that resulting parameter estimates could be interpreted as representing an example 76.5-year-old female with 17.4 years of education.

*Figure 1* shows the slopes of PIB DVR and CVLT over time for each subject. PIB DVR slopes across time (relative to baseline MRI) for each subject are shown in *Figure 1a*. CVLT slopes across time (relative to baseline MRI) are shown in *Figure 1b*. The average time from the baseline MRI to last PIB scan was 3.4 years (range 2.8–6 years) and from MRI to last neuropsychological session was 2.7 years (range 1.5–6.1 years). Participants came in for PIB scans and neuropsychological testing sessions on separate days. The average time between MRI and first PIB scan was about one week and the average time between MRI and first neuropsychological session was about eight weeks. At the baseline exam, six subjects were classified as PIB+ (four were ApoE4 carriers) and by the end of the study eight subjects were PIB+ (both who converted to PIB+ were ApoE4 carriers).

## Hippocampal activation at baseline associated with longitudinal Aβ accumulation

To determine whether hippocampal activation at baseline predicted longitudinal accumulation of global Aβ, we examined associations between our independent variables of interest (hippocampal activation and time of PIB) and the dependent variable (PIB DVR) using a linear mixed model (with age, sex, and education as covariates). Time of PIB was determined as relative to baseline MRI. The interaction between hippocampal activation and time of PIB (hippocampal activation × time) was the primary effect of interest because it represented the relationship between hippocampal activation and longitudinal PIB DVR change. Results revealed that hippocampal activation at baseline was positively associated with longitudinal amyloid accumulation such that PIB DVR increased at a rate of 0.001 units per year for every one hippocampal activation unit [interaction, p=0.004, *Table 2*]. To visualize this, we plotted the relationship between PIB slope from the model and hippocampal activation contrast values at baseline (*Figure 2a*), which showed a positive correlation between PIB slope and hippocampal activation.

We also performed the same linear mixed model described above, but with non-hippocampal activation as an independent variable to determine if brain activation predicting longitudinal Aβ accumulation was specific to the hippocampus or existed for cortical activity as well. We chose the

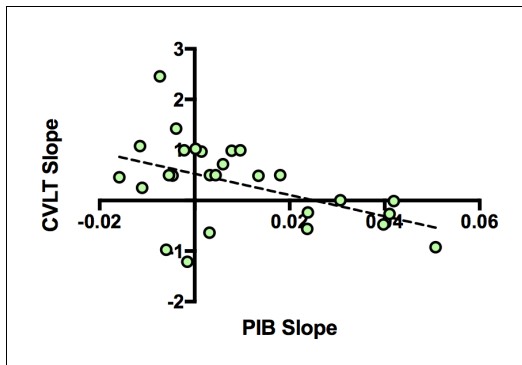

**Figure 3.** Relationship between longitudinal amyloid accumulation and memory decline. Negative relationship between [11C] Pittsburgh Compound-B distribution volume ratio slopes (PIB slope) and California Verbal Learning Test Long-Delay Free Recall slope (CVLT slope measured as CVLT scores over time) obtained from the linear mixed model.

non-hippocampal regions of bilateral occipital cortex and bilateral inferior frontal gyrus, which similarly to the hippocampus, showed increased activity in PIB+ compared to PIB- subjects in the original study (*Mormino et al., 2012*). However, we did not find an interaction between non-hippocampal cortical activation and longitudinal A$\beta$ accumulation [interaction, t(21) = 1.02, p=0.319; *Figure 2b*], suggesting increased brain activation at baseline associated with increased global A$\beta$ accumulation may be MTL specific.

To determine whether hippocampal activation at baseline was associated with change in memory performance over time, we examined associations between our independent variables of interest (hippocampal activation and time of CVLT) and the dependent variable (CVLT) using a linear mixed model. As with time of PIB, time of CVLT was determined as relative to baseline MRI. The interaction between hippocampal activation and time of CVLT (hippocampal activation × time) was not significant [interaction, t(14) = 0.786, p=0.445] indicating no relationship between hippocampal activation at baseline and the subsequent rate of memory decline.

## Increased A$\beta$ accumulation associated with a longitudinal decline in memory performance

Next, we examined the relationship between longitudinal A$\beta$ accumulation and memory performance. We examined associations between the independent variables of interest (PIB slope and time of CVLT) and the dependent variable (CVLT) using a linear mixed model. The interaction between PIB slope and time of CVLT (PIB slope × time) was the primary effect of interest because it represented the relationship between A$\beta$ accumulation and memory change over time. Results revealed that increased longitudinal A$\beta$ accumulation was associated with a decline in CVLT score [interaction, p=0.023, *Table 3*]. To visualize the effect, we plotted the relationship between CVLT slope from the model and PIB slope (*Figure 3*), which showed a negative correlation between CVLT slope and PIB slope. Sex [p = 0.034] and time of CVLT [p = 0.023] also predicted the average CVLT score (*Table 3*). The effect of time of CVLT can be seen in *Figure 1b*, where the average CVLT slope increases over time, by 0.41 CVLT units/year. This suggests that while many participants retain their memory abilities over time, longitudinal A$\beta$ accumulation predicts memory decline over time.

**Table 2.** Linear mixed model results for hippocampal activation predicting longitudinal amyloid accumulation.

| Parameter | Estimate (SE) | df | t | p |
|---|---|---|---|---|
| Intercept | 1.00 (0.07) | 22 | 14.58 | <0.001 |
| Age (at MRI, mc) | −0.001 (0.01) | 22 | −0.14 | 0.887 |
| Sex (F) | 0.10 (0.08) | 22 | 1.27 | 0.216 |
| Education (years, mc) | −0.02 (0.02) | 22 | −0.89 | 0.385 |
| Hippocampal activation (baseline) | 0.004 (0.002) | 22 | 1.64 | 0.115 |
| Time (PIB DVR) | 0.003 (0.003) | 12 | 0.87 | 0.399 |
| Hippocampal activation * Time (PIB DVR) | 0.001 (0.0002) | 12 | 3.58 | **0.004** |

Dependent Variable: PIB DVR.

*Key:* SE = Standard Error, mc = mean-centered, PIB DVR= Pittsburgh Compound B Distributed Volume Ratio.

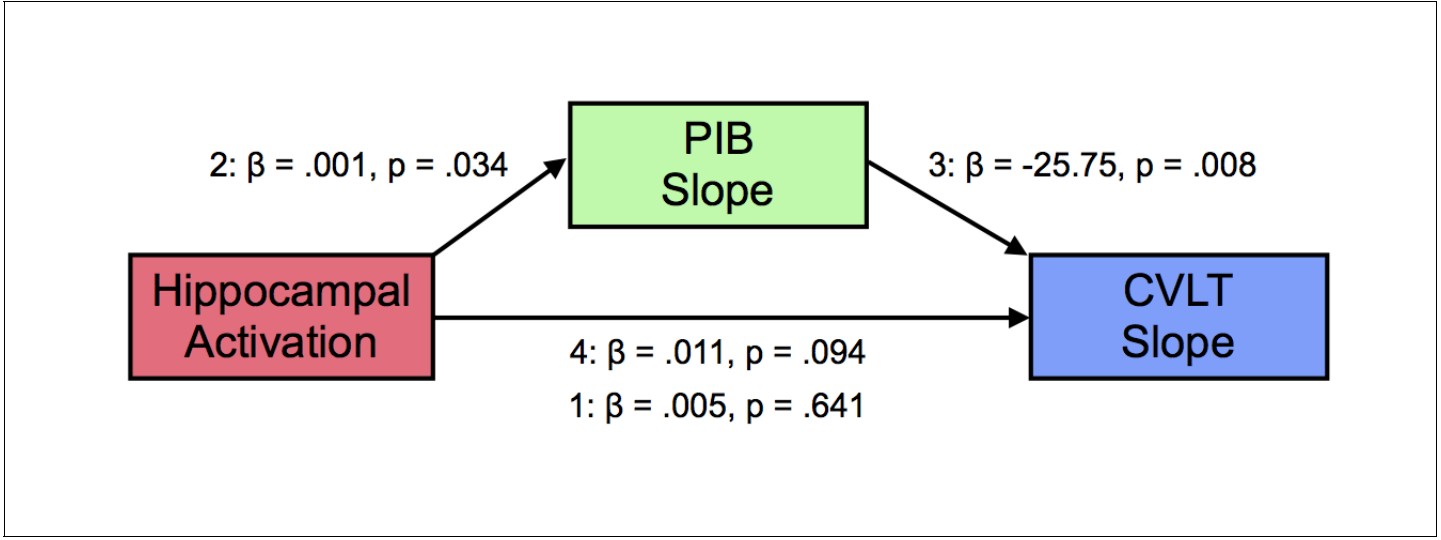

**Figure 4.** Mediation analysis between hippocampal activation, longitudinal amyloid accumulation, and memory decline. Mediation analysis with hippocampal activation at baseline (predictor, X), mediated by [11C] Pittsburgh Compound-B distribution volume ratio slopes (PIB Slope) (mediator, M), predicting California Verbal Learning Test Long-Delay Free Recall slope (CVLT Slope). Hippocampal activation positively predicted PIB Slope, PIB Slope negatively predicted CVLT Slope, which resulted in a significant mediation (see text for more details).

### Longitudinal amyloid accumulation mediates the influence of hippocampal activation on memory decline

Based on the observation that hippocampal activation at baseline was related to increasing A$\beta$ deposition over time, we conjectured that the relationships between the three variables in this study were complex. Thus, we examined whether the increased longitudinal A$\beta$ deposition associated with baseline hippocampal activation was linked to memory decline. We tested this relationship using a mediation model, where PIB slope (mediator, M) may mediate the relationship between hippocampal activation at baseline (predictor, X) and CVLT slope (outcome, Y), with age, sex, and education as covariates. Step 1 tested the effect of X on Y, not including M. Step 2 tested X predicting M, and Steps 3 and 4 tested M|X significant predictor of Y and X|M not a significant predictor of Y. In Step 1 of the model, the regression of hippocampal activation on memory performance, ignoring PIB slope, was not significant [$\beta = 0.005$, $t = 0.473$, $p=0.641$]. In Step 2, the regression of hippocampal activation on PIB slope was significant [$\beta = 0.0005$, $t = 2.26$, $p=0.034$]. In Step 3, the regression of PIB slope, controlling for hippocampal activation, on memory performance was significant [$\beta = -25.75$,

**Table 3.** Linear mixed model results for longitudinal amyloid accumulation predicting memory decline.

| Parameter | Estimate (SE) | df | t | p |
|---|---|---|---|---|
| Intercept | 10.07 (0.89) | 18 | 11.27 | **<0.001** |
| Age (at MRI, mc) | −0.06 (0.09) | 16 | −0.69 | 0.499 |
| Sex (F) | 2.54 (1.10) | 16 | 2.31 | **0.034** |
| Education (years, mc) | −0.17 (0.28) | 17 | 0.59 | 0.562 |
| Slope (PIB DVR) | 15.6 (28.32) | 21 | 0.55 | 0.587 |
| Time (CVLT LD FR) | 0.41 (0.17) | 23 | 2.44 | **0.023** |
| Slope (PIB DVR) * Time (CVLT LD FR) | −20.06 (8.26) | 25 | −2.43 | **0.023** |

Dependent Variable: CVLT LD FR.

*Key:* SE = Standard Error, mc = mean-centered, PIB DVR= Pittsburgh Compound B Distributed Volume Ratio, CVLT LD FR = California Verbal Learning Test - Long Delay Free Recall.

t = −2.91, p=0.008]. In Step 4, when controlling for PIB slope, hippocampal activation was not a significant predictor of memory performance [$\beta$ = 0.018, t = 1.76, p=0.094]. There were no significant effects of age, sex, or education. The bootstrapped unstandardized indirect effect was −0.016 and significantly differed from zero, as revealed by a 95% bootstrap confidence interval that was entirely below zero (confidence interval: −0.031 to −0.002). Thus, longitudinal amyloid accumulation mediated the relationship between increased baseline hippocampal activation and greater memory decline over time (*Figure 4*).

Since prior studies have noted increased A$\beta$ accumulation over time in those with elevated baseline A$\beta$ (*Villemagne et al., 2011*; *Sojkova et al., 2011*) and because we previously found a relationship between baseline A$\beta$ and activation (*Mormino et al., 2012*), we analyzed the relationship between baseline PIB, hippocampal activation, and PIB slope to ensure that our effects on longitudinal A$\beta$ accumulation were not being driven solely by elevated baseline A$\beta$. To determine whether increased baseline amyloid levels led to increased baseline hippocampal activation, which led to greater longitudinal A$\beta$ accumulation, we conducted a mediation analysis, with age, sex, and education as covariates. We found no evidence of mediation, as revealed by a 95% bootstrap confidence interval that included zero (confidence interval: −0.004 to 0.044). Thus, baseline amyloid levels did not lead to greater baseline hippocampal activation to influence longitudinal A$\beta$ accumulation, suggesting the effect of hippocampal activation on A$\beta$ accumulation does not solely rely on baseline amyloid levels. Furthermore, by including random intercepts into our linear mixed models, we accounted for initial differences in amyloid. We then conducted a moderation analysis using hierarchical linear regression to determine if baseline amyloid levels influenced A$\beta$ accumulation, but depended on the level of hippocampal activation. The two predictors (hippocampal activation and baseline PIB) were first entered into the regression analysis to determine each predictor's effect on PIB slope and then the interaction term was added. Results indicated that baseline hippocampal activation [b = −0.004, p=0.034] but not baseline PIB [b = 2.01, p=0.608] was associated with PIB slope and accounted for a significant amount of variance in PIB slope [$R^2$ = 0.524, F(6,20) = 3.67, p=0.013]. The interaction between hippocampal activation and baseline PIB explained a significant increase in variance in PIB slope [$\Delta R^2$ = 0.15, F(1,20) = 6.32, p=0.021]. Thus, the influence of baseline amyloid levels on longitudinal A$\beta$ accumulation depends on hippocampal activation at baseline. To further characterize the nature of this interaction, we used the Johnson-Neyman technique to identify at which point in the range of hippocampal activation contrast values where the effect of the predictor on the outcome transitions from being statistically significant to non-significant (*Hayes, 2013*). We found that when hippocampal activation was greater than or equal to a contrast value of 12.18, higher baseline amyloid led to greater A$\beta$ accumulation. This suggests that while baseline amyloid levels do not directly lead to greater levels of hippocampal activation, baseline amyloid's relationship with A$\beta$ accumulation may depend on higher levels of hippocampal activation at baseline.

## Discussion

In this study, we used fMRI with PIB-PET imaging and neuropsychological testing to investigate the influence of memory-related hippocampal activation at baseline on longitudinal A$\beta$ accumulation and cognitive decline in normal older adults. While the amyloid hypothesis places A$\beta$ deposition as the earliest event in a pathological cascade, why A$\beta$ accumulates in the first place is not known. Although A$\beta$ can induce structural and functional alterations to synapses and synaptic plasticity (*Selkoe, 2002*; *Almeida et al., 2005*; *Coleman and Yao, 2003*), the reverse relationship also exists such that early synaptic activity can influence A$\beta$ (*Nitsch et al., 1993*; *Kamenetz et al., 2003*; *Cirrito et al., 2005*; *Bero et al., 2011*; *Brody et al., 2008*). A recent mouse model of AD found that hyperactivity of hippocampal neurons precedes A$\beta$ plaque formation, suggesting that hippocampal hyperactivity is one of the earliest dysfunctions in the pathophysiological cascade of AD (*Busche and Konnerth, 2016*). This indicates that a major target of A$\beta$ itself, synaptic activation, can control A$\beta$ levels.

These results support the hypothesis that increased hippocampal activation at baseline is associated with increased longitudinal A$\beta$ accumulation across frontal, parietal, temporal, and cingulate cortices and that longitudinal A$\beta$ accumulation mediates the effect of baseline hippocampal activation on memory decline. Although previous studies have shown that baseline PIB is associated with the rate of longitudinal amyloid accumulation (*Villemagne et al., 2011*; *Sojkova et al., 2011*), our

results are not simply dependent on baseline PIB for several reasons. First, we included random slopes and random intercepts in our linear mixed models to account for baseline PIB and different PIB slopes for each participant. Second, we performed a mediation analysis to determine if baseline PIB acted through baseline hippocampal activation to lead to increased longitudinal A$\beta$ accumulation, which was revealed not to be the case. Finally, a moderation analysis showed that baseline PIB predicted increased longitudinal A$\beta$ accumulation, but only when baseline hippocampal activation was high. These data suggest that while baseline amyloid may be linked to increased A$\beta$ accumulation, this did not drive the relationships between baseline hippocampal activity and longitudinal A$\beta$ accumulation.

Previous studies have described increased brain activity in those at risk for AD in different regions: the dentate gyrus/CA3 region of the hippocampus (*Bakker et al., 2012*; *Yassa, 2011*), the entire hippocampus (*Mormino et al., 2012*; *Dickerson et al., 2005*), the MTL more broadly (*Dickerson et al., 2004*), or neocortex (*Mormino et al., 2012*; *Elman et al., 2014*; *Jones et al., 2016*; *Myers et al., 2014*). These studies suggest that increased activation during memory performance is linked to risk for AD in older adults with memory impairment, MCI, ApoE4+, or those who are A$\beta$+. Our data support a role for task-related brain activation in the hippocampus that is linked to global A$\beta$ deposition. Indeed, we found that only hippocampal activation at baseline and not non-hippocampal cortical activation was linked to increased longitudinal A$\beta$ accumulation, suggesting regional specificity in these relationships. The regional pattern of relationships between cortical activation, hippocampal activation, and AD-related pathology may appear perplexing. Cross-sectional neuropathological data suggest that A$\beta$ is first deposited in the cortex and then spreads to the MTL (*Thal et al., 2002*), while tau deposition in the form of neurofibrillary tangles deposit in the MTL initially with later spread to neocortex (*Braak and Braak, 1995*). We find that activation in the hippocampus predicts cortical A$\beta$ while cortical activation does not. There are a few possible explanations for this. Hippocampal hyperactivity may be more indicative of pathological effects in those at risk for AD, while cortical activation may reflect other processes. For example, we previously reported that increased cortical activity, but not hippocampal activity appeared to be compensatory during memory encoding (*Elman et al., 2014*). This is consistent with data associating increased hippocampal activation with memory deficits (*Yassa et al., 2011*), and with the findings that reversing hippocampal hyperactivity rescues memory deficits (*Bakker et al., 2012*, *2015*). Another possibility is that increased hippocampal activation is driving neuronal activity elsewhere, in cortical regions such as retrosplenial cortex and precuneus with strong hippocampal connectivity and early A$\beta$ deposition. Another possibility is that hippocampal activation is driving tau deposition in the MTL, which then leads to A$\beta$ deposition. In fact, preclinical data suggest a crucial role for tau accumulation in producing aberrant neural hyperactivity (*Roberson et al., 2011*). These possibilities are not mutually exclusive and will be important to examine in the future.

This study has several limitations. Not all subjects had the same number of PIB scans (each subject had two or three scans), which may influence the accuracy of PIB slope, however, linear mixed models are equipped to handle missing data. We found that there was no significant difference in PIB slope between participants with two or three scans or a relationship between years followed and PIB slope (p's > 0.05). Sample size was relatively small and we did not have a large enough number of subjects to fully examine sex or ApoE4 carrier differences adequately. Absence of association may be due to lack of power rather than absence of a relationship, such as the absence of ApoE4 effect and absence of mediation through baseline A$\beta$. However, to determine if the effects of hippocampal activity on amyloid accumulation was driven by ApoE4 carriers, we removed them from the sample and re-ran the linear mixed model only in ApoE4 non-carriers. We found the interaction between hippocampal activation and amyloid accumulation remained significant [p = 0.018]. Thus, it appears the effect of hippocampal activation on increased amyloid accumulation is not specific to those who are ApoE4 carriers, but is also occurring in ApoE4 non-carriers that are also accumulating amyloid.

Another limitation of the study ignores the role of tau in producing these effects. As tau PET imaging is still relatively novel, future studies can incorporate tau into these models to determine if hippocampal activation at baseline influences longitudinal A$\beta$ accumulation alone or if it influences how A$\beta$ and tau interact during the progression of AD. It will also be important to include longitudinal fMRI data in addition to longitudinal PIB-PET data to measure how longitudinal change in hippocampal activation tracks longitudinal A$\beta$ accumulation. One would expect that as hippocampal activation decreases over time (*Celone et al., 2006*; *O'Brien et al., 2004*), increases in A$\beta$

accumulation will taper off, as this would indicate later stages of AD. In fact, data suggest that late in disease stage, Aβ deposition slows or even reverses, consistent with the idea that declining brain activity affects late-stage Aβ deposition (*Villemagne et al., 2011*; *Jack et al., 2013*). While we find that baseline amyloid levels did not drive our findings of hippocampal activation in predicting Aβ accumulation, it will be important for future studies to track those with little baseline amyloid deposition to determine if they eventually display a pattern of increased hippocampal activity followed by increased amyloid deposition.

## Materials and methods

### Participants

Forty-five normal older adults (N = 45) underwent PIB-PET imaging, fMRI, and neuropsychological testing for this study. Subjects participated in an fMRI experiment of memory, the results of which have been published (*Mormino et al., 2012*). From this sample, we included subjects who had (1) participated in the fMRI experiment, (2) at least two PIB scans, and (3) three neuropsychological testing sessions. All met the following inclusion criteria: no MRI contradictions, living independently in the community, MMSE ≥26, within age, education, and gender norms on cognitive tests, absence of neurological or psychiatric illness, and lack of major medical illnesses and medications that affect cognition. Fifteen of the 45 subjects did not return for follow-up PIB-PET imaging, so they were not included in the current study. Three additional subjects were excluded due to problems with their PIB-PET data (one had too much gray matter atrophy to accurately measure PIB, one had incomplete acquisition on their follow-up PIB scan, and one was greater than 3.5 standard deviations above the mean for non-hippocampal activation). This left us with a sample of 27 subjects who met the criteria for the study (see *Table 1* for demographics).

### Neuropsychological testing

The California Verbal Learning Test was administered to each participant as part of a larger battery of tests used at an annual evaluation carried out by the Berkeley Aging Cohort Study (BACS). Participants received a neuropsychological evaluation within three months of participating in the fMRI study. Every subject had three neuropsychological testing sessions included for this study (average time since MRI = 2.7 years). The neuropsychological battery was designed to examine memory function, as well as other aspects of general cognitive ability. The CVLT is a standardized memory test that was developed to assess a variety of memory processes and assesses rate of learning, retention after short- and long-delay intervals, semantic encoding ability, and recognition memory. We chose to focus on the CVLT Long-Delay Free Recall component of the test (out of 16), as this has been shown to be sensitive to cognitive decline and episodic memory deficits with age (*Lange et al., 2002*; *Delis et al., 1991*).

### Apolipoprotein E genotyping

Participants' DNA from blood samples was analyzed for apolipoprotein E (ApoE) polymorphisms using a standard protocol. For statistical comparison between groups, subjects were dichotomized into carriers and non-carriers of the E4 allele (see *Table 1* for more details).

### PIB-PET acquisition

PIB was synthesized at the Lawrence Berkeley National Laboratory's (LBNL) Biomedical Isotope Facility using a published protocol and described in detail previously (*Mormino et al., 2012*; *Mathis et al., 2003*). PIB-PET imaging was performed at LBNL using an ECAT EXACT HR or BIOGRAPH Truepoint six scanner (Siemens Medical Systems, Erlangen, Germany) in three dimensional acquisition mode. Ten to fifteen mCi of PIB was injected into an antecubital vein. Dynamic acquisition frames were obtained as follows: 4 × 15, 8 × 30, 9 × 60, 2 × 180, 8 × 300, and 3 × 600 s (90 min total). Ten-minute transmission scans for attenuation correction or X-ray CT were obtained for each PIB scan. Data were corrected for motion and reconstructed with an iterative ordered subset expectation maximization algorithm with weighted attenuation. Images were smoothed with a 4 mm Gaussian kernel with scatter correction.

## PIB-PET processing

All PIB-PET data were preprocessed using SPM12 software (http://www.fil.ion.ucl.ac.uk/spm/). The first five minutes of data were summed and then a two-pass realignment was performed. All scans were realigned to the summed first five minutes followed and then created of a mean of all those images followed by realigning everything to that mean. PET images were then co-registered to each subject's structural MRI (which were repeated near the time of each PET scan). PIB DVR images were created using Logan graphical analysis with frames corresponding to 35–90 min after injection and a cerebellar gray matter reference region defined using FreeSurfer version 5.3. Mean DVR values from frontal, parietal, temporal, and cingulate cortices were computed to serve as a global PIB index for all subjects. All subjects were scanned on the ECAT HR for their first PIB scan, four subjects were scanned on the BIOGRAPH for their second scan (the rest on the ECAT HR) and eight subjects were scanned on the BIOGRAPH if they had a third PIB scan. There were no significant effects of scanner type in any of the linear mixed models and we have previously shown PIB measurement on these different scanners to be equivalent (*Elman et al., 2014*).

## MRI acquisition

The following MRI acquisition parameters were previously reported (*Mormino et al., 2012*). All subjects underwent MRI scanning at LBNL on a 1.5T Magnetom Avanto System (Siemens Medical Systems) with a 12-channel head coil run in triple mode. A high-resolution structural T1-weighted volumetric magnetization prepared rapid gradient echo scan (MP-RAGE, axially acquired, time repetition [TR]/time echo [TE]/time to inversion [TI] = 2110/3.58/1100 ms, flip angle = 15°, 1.00 × 1.00 mm$^2$ in plane resolution, 1.00 mm thickness with 50% gap) and a low-resolution structural T1-weighted in plane to the fMRI scans were collected (axially acquired, TR/TE = 591/10/10 ms, flip angle = 150°, 0.90 × 0.90 mm$^2$ in plane resolution, 3.40 mm thickness with 15% gap). For fMRI scanning, 4 $T_2^*$-weighted gradient-echo echo planar images (EPI) were collected (28 axially acquired slices, TR/TE = 2200/50 ms, flip angle = 90°, 3.40 × 3.40 mm$^2$ in plane resolution, 3.40 mm thickness with 15% gap).

## fMRI episodic memory task

Two hundred images of natural outdoor scenes were presented for 4.4 s each. Subjects were instructed to indicate whether water was present in each image. Scans were broken into four sessions, with 50 scenes and 185 TRs per session. Zero to 5 TRs of fixation (green crosshair on black background) were randomly intermixed between scenes to allow separation of individual trials (average inter-stimulus interval = 3.46 s). After the scan, a surprise recognition task including all stimuli presented during encoding as well as 100 novel stimuli (foils) was used to assess performance and sort fMRI data. There was a 15 min delay between the last stimulus encoded and the start of the memory test. For each image, subjects were asked if they had seen the image before and were allowed to respond with 1 of 4 responses: high-confidence yes, low-confidence yes, high-confidence no, and low-confidence no. The task was self-paced and subjects were encouraged to be as accurate as possible. For the current study, we examined task activation (mean contrast values: hits > misses) from task-positive regions of interest (ROIs), which included the right hippocampus, bilateral occipital cortex, and bilateral inferior frontal gyrus, as described in *Mormino et al. (2012)*. MP-RAGE scans were processed using FreeSurfer version 5.3 (https://surfer.nmr.mgh.harvard.edu) to derive ROIs in each subject's native space. Detailed task results and fMRI processing have been reported previously (*Mormino et al., 2012*).

## Statistical analyses

All statistical analyses of behavioral variables and ROI activation means were conducted in SPSS v. 24 (IBM Corp., Armonk, NY). We conducted a series of linear mixed models. All models included age, sex, and education as covariates, as well as a random intercept to account for individual variability in initial PIB DVR or CVLT scores and random slopes to account for individual variability in PIB DVR or CVLT slopes. We also employed an autoregressive covariance structure to account for correlations between consecutive PIB DVR or CVLT values over time. Initial models also included ApoE4 carrier (+/−) and scanner type (ECAT HR, BIOGRAPH) as covariates, but effects were not significant so we removed them to simplify the model. Age was mean-centered at 76.5 years, education was

mean-centered at 17.4 years, and females were the reference group, so that resulting parameter estimates could be interpreted as representing an example 76.5-year-old female with 17.4 years of education. While PIB and CVLT were both time-varying measurements, we did not have the statistical power to include both as time-varying measurements in our mixed effects model testing change in PIB predicting change in memory performance over time, since visits did not coincide in many cases, making modeling with a single time challenging. We therefore used PIB slope as an alternative, but note that this is a limitation as it does not precisely capture PIB change relative to CVLT change.

Mediation and moderation analyses were conducted in SPSS using the PROCESS module (*Hayes, 2013*). For mediation analysis, unstandardized indirect effects were computed for each of 1000 bootstrapped samples, and the 95% confidence interval was computed. For the mediation model, Step 1 tested the effect of X on Y, not including M. Step 2 tested X predicting M, and Steps 3 and 4 tested M|X significant predictor of Y and X|M not a significant predictor of Y. For moderation analysis, the module employs hierarchical regression analysis to first test the relationships between the predictor and the outcome and the moderator and the outcome (conditional effects) and to then test the interaction between the predictor and the moderator (i.e. how the effect of the predictor on the outcome changes as the moderator changes). Furthermore, it provides data to probe the interaction further by determining where in the distribution of the moderator the predictor is related to the outcome to better discern the interpretation of the interaction (i.e. Johnson-Neyman technique). Statistical values were considered significant at a final alpha level of. 05 to prevent Type I error inflation.

## Acknowledgements

We thank Sam Lockhart, Anne Maass, and Michael Scholl for helpful discussions, and Suzanne Baker, Elizabeth Mormino, Taylor Mellinger, and Kaitlin Swinnerton for their assistance with data collection and processing. Research reported in this publication was supported by the National Institute of Aging of the National Institutes of Health under Award Number F32AG054116 and by AG034570.

## Additional information

### Funding

| Funder | Grant reference number | Author |
| --- | --- | --- |
| National Institute on Aging | AG054116 | Stephanie L Leal |
| National Institute on Aging | AG034570 | William J Jagust |

The funders had no role in study design, data collection and interpretation, or the decision to submit the work for publication.

### Author contributions

SLL, Conceptualization, Formal analysis, Writing—original draft, Writing—review and editing; SML, Formal analysis, Writing—review and editing; RKB, Data curation, Formal analysis, Writing—review and editing; WJJ, Conceptualization, Formal analysis, Funding acquisition, Writing—review and editing

### Author ORCIDs

Stephanie L Leal, http://orcid.org/0000-0002-8082-8291

### Ethics

Human subjects: Informed consent was obtained from all research participants and approved by the Institutional Review Boards of Lawrence Berkeley National Labs and UC Berkeley.

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
