## [Decision Letter]

Thank you for submitting your article "Hippocampal activation is associated with longitudinal amyloid accumulation and cognitive decline" for consideration by *eLife*. Your article has been favorably evaluated by Gary Westbrook (Senior Editor) and three reviewers, one of whom is a member of our Board of Reviewing Editors. The following individual involved in review of your submission has agreed to reveal their identity: Michela Gallagher (Reviewer #3). The reviewers have discussed the reviews with one another and the Reviewing Editor has drafted this decision to help you prepare a revised submission.

Summary:

Leal and colleagues investigated the connection between neuronal activity and amyloid-β (Aβ) in the brain of cognitively normal humans. Specifically, they assessed hippocampal activity using fMRI during a memory task which included encoding and recall of visual images, and determined how this baseline hippocampal activity was associated with amyloid deposition over time, which was measured by 11C-Pittsburgh Compound-B (PIB) PET neuroimaging. Long-term memory changes were also measured by a verbal learning long-delay free recall test (CVLT). The authors report a significant, positive association between baseline hippocampal activation and longitudinal PIB changes. This association appeared to be specific to the hippocampus, as occipital cortex and bilateral inferior frontal gyrus did not show any association with PIB change over time. There was also no association between hippocampal activity and longitudinal memory changes, but longitudinal amyloid accumulation mediated the relationship between these two variables. There was also an association between Aβ accumulation and CVLT decline over time. The authors point out that they have previously found baseline alterations in brain activity due to the presence of amyloid plaques, but they did not see a connection between PIB levels and hippocampal activity at baseline suggesting that hippocampal activity was influencing Aβ deposition independent of amyloid levels at baseline.

Essential revisions:

1) The authors point out that due to their smaller sample size, they were unable to examine the specific effects of ApoE4 carriers. This is understandable, but of the 6-7 participants who were PIB +, how many of those were ApoE4 carriers? This information should be reported. If all of the PIB + individuals were ApoE4 carriers, the effects of hippocampal activity on amyloid accumulation could be mediated by ApoE4 and it would be unclear what the relationship between hippocampal activity and amyloid accumulation is in non-ApoE4 individuals. Second, because we know APOE4 is associated with increased Aß deposition it should be included as a covariate in all of the analyses including PIB measures.

2) PIB distribution volume ratios were calculated from reactivity in cortical regions: frontal, parietal, temporal, and cingulate cortex. This allowed the effects of hippocampal activity on global amyloid deposition to be assessed. However, results from mouse studies would suggest that Aβ levels would also be increased locally, in the hippocampus, with higher baseline hippocampal activity (e.g. Cirrito et al., 2005). Was there any amyloid accumulation in the hippocampus of these subjects that was detectable and if so, does longitudinal hippocampal PIB also associate with baseline hippocampal activity?

3) Subsection “Increased Aβ accumulation associated with a longitudinal decline in memory performance”: The choice of the model here could be of some concern. The authors might consider that both PIB and CVLT are time-varying outcomes. They have summarized the changes in PIB over time and used this slope as a predictor of changes in CVLT over time. So in essence this approach uses data from PIB that occurs after a CVLT measurement to inform what will happen to CVLT at that time. Isn't this logic a bit circular?

The authors might consider an alternative modeling approach and see whether it yields the same results. Consider:

CVLT _ij = b0 + b1 PIB_i0 + b2 (PIB_ij – PIB_i0) + b3 time_ij + b4 (PIB_ij – PIB_i0) x time_ij

Basically, consider how changes from baseline in PIB are associated with changes at CVLT BUT only up to the particular time point. They can add in their random intercept and random slope for time as they did in the models they fit.

4) There is a similar concern with the analysis for amyloid accumulation mediating the influence of hippocampal activation on memory decline, where they have done the analysis on the summary measures of slope (subsection “Longitudinal amyloid accumulation mediates the influence of hippocampal activation on memory decline”).

Step 1 tested the effect of X on Y, not including M. -> Likely this is fine.

Step 2 tested X predicting M -> this is basically the model they report in the subsection “Hippocampal activation at baseline associated with longitudinal Aβ accumulation”.

Steps 3 and 4 tested M|X significant predictor of Y and X|M not a significant predictor of Y -> Here they could also adapt the model suggested above to include X.

---

## [Author Response]

*Essential revisions:*

*1) The authors point out that due to their smaller sample size, they were unable to examine the specific effects of ApoE4 carriers. This is understandable, but of the 6-7 participants who were PIB +, how many of those were ApoE4 carriers? This information should be reported. If all of the PIB + individuals were ApoE4 carriers, the effects of hippocampal activity on amyloid accumulation could be mediated by ApoE4 and it would be unclear what the relationship between hippocampal activity and amyloid accumulation is in non-ApoE4 individuals. Second, because we know APOE4 is associated with increased Aß deposition it should be included as a covariate in all of the analyses including PIB measures.*

There were 6 PIB+ subjects at baseline, 4 of whom were ApoE4 carriers. Over time, 2 additional subjects converted to PIB+; both were ApoE4 carriers. This is now added to the manuscript.

Since most subjects were ApoE4 non-carriers we had a reasonable sample size to re-run the linear mixed model of hippocampal activity predicting Aβ accumulation only in the ApoE4 non-carriers and found the interaction between hippocampal activation and amyloid over time remained significant [p =.018]. Thus, the effect of hippocampal activation on increased amyloid accumulation is not specific to those who are ApoE4 carriers, but is also occurring in ApoE4 non-carriers who are also accumulating amyloid. We added this analysis to the Discussion: “However, to determine if the effects of hippocampal activity on Aβ accumulation was driven by ApoE4 carriers, we removed them from the sample and re-ran the linear mixed model only in ApoE4 non-carriers. We found the interaction between hippocampal activation and Aβ accumulation remained significant [p =.018]. Thus, it appears the effect of hippocampal activation on increased Aβ accumulation is not specific to those who are ApoE4 carriers, but is also occurring in ApoE4 non-carriers who are also accumulating Aβ.”

As stated in the Statistical analyses section of the Methods, “Initial models included ApoE4 carrier status (+/-) and scanner type (ECAT HR, BIOGRAPH) as covariates, but effects were not significant so we removed them to simplify the model.”

*2) PIB distribution volume ratios were calculated from reactivity in cortical regions: frontal, parietal, temporal, and cingulate cortex. This allowed the effects of hippocampal activity on global amyloid deposition to be assessed. However, results from mouse studies would suggest that Aβ levels would also be increased locally, in the hippocampus, with higher baseline hippocampal activity (e.g. Cirrito et al., 2005). Was there any amyloid accumulation in the hippocampus of these subjects that was detectable and if so, does longitudinal hippocampal PIB also associate with baseline hippocampal activity?*

We ran a linear mixed model to determine if local hippocampal Aβ accumulation was associated with increased hippocampal activation, with age, sex, and education as covariates. We found a marginal interaction [p=.056] between baseline hippocampal activation and hippocampal amyloid accumulation over time, suggesting that in addition to the effect of hippocampal activation on global amyloid accumulation, there appears to be a local relationship as well, although not quite significant. Aβ first accumulates in the neocortex in phase 1, followed by medial temporal lobe regions including the hippocampus in phases 2-3 (Thal et al., 2002). This may explain why the relationship is stronger for global amyloid measures rather than local hippocampal amyloid measures, especially given that our subject population is comprised of healthy cognitively normal older adults. Furthermore, the hippocampus is a brain region where Aβ is difficult to measure, because its small volume and adjacent white matter and CSF makes quantification problematic. Because of these issues, PET studies rarely report hippocampal Aβ accumulation.

3) Subsection “Increased Aβ accumulation associated with a longitudinal decline in memory performance”: The choice of the model here could be of some concern. The authors might consider that both PIB and CVLT are time-varying outcomes. They have summarized the changes in PIB over time and used this slope as a predictor of changes in CVLT over time. So in essence this approach uses data from PIB that occurs after a CVLT measurement to inform what will happen to CVLT at that time. Isn't this logic a bit circular?

*The authors might consider an alternative modeling approach and see whether it yields the same results. Consider:*

*CVLT _ij = b0 + b1 PIB_i0 + b2 (PIB_ij – PIB_i0) + b3 time_ij + b4 (PIB_ij – PIB_i0) x time_ij*

*Basically, consider how changes from baseline in PIB are associated with changes at CVLT BUT only up to the particular time point. They can add in their random intercept and random slope for time as they did in the models they fit.*

We understand the reviewers’ concern that the use of PIB slope results in loss of information because of the time-varying nature of PIB. A related methodological point that may not have been clear is that PIB scans and CVLT testing sessions were not done on the same day (and sometimes occurred over a year from one another). The non-overlapping timing of the PIB & CVLT visits made it difficult to use a single time variable to account for both time-varying PIB *and* CVLT measurements as suggested in the model above.

Nonetheless, we attempted an alternative mixed effects model in which we fit a model with both time-varying CVLT and PIB measurements and a collapsed time measure in units of years since the baseline MRI, along with the previous covariates and random slope/intercept. For the time variable, we matched CVLT and PIB measurements in time where visits overlapped, but many visits had only CVLT *or* PIB, resulting in numerous missing values over time for each variable. This model did not converge on a fit, as it did not have enough statistical power for the full formal random effects model with 27 subjects. Therefore, our original analysis, the use of PIB slope to estimate PIB change for each subject, was our best option given the current sample size and number of longitudinal time points.

However, we have noted this limitation in the Statistical Analyses section in the Methods: “While PIB and CVLT were both time-varying measurements, we did not have the statistical power to include both as time-varying measurements in our mixed effects model testing change in PIB predicting change in memory performance over time, since visits did not coincide in many cases, making modeling with a single time challenging. We therefore used PIB slope as an alternative, but note that this is a limitation as it does not precisely capture PIB change relative to CVLT change.”

4) There is a similar concern with the analysis for amyloid accumulation mediating the influence of hippocampal activation on memory decline, where they have done the analysis on the summary measures of slope (subsection “Longitudinal amyloid accumulation mediates the influence of hippocampal activation on memory decline”).

Step 1 tested the effect of X on Y, not including M. -> Likely this is fine.

Step 2 tested X predicting M -> this is basically the model they report in the subsection “Hippocampal activation at baseline associated with longitudinal Aβ accumulation”.

*Steps 3 and 4 tested M|X significant predictor of Y and X|M not a significant predictor of Y -> Here they could also adapt the model suggested above to include* X.

As we state above, since our CVLT and PIB measures were not completed at the same visits, it makes it difficult with a small sample to predict on a time point by time point basis.